# FedPeWS: Personalized Warmup via Subnetworks for Enhanced Heterogeneous Federated Learning

## Abstract

Statistical data heterogeneity is a significant barrier to convergence in federated learning (FL). While prior work has advanced heterogeneous FL through better optimization objectives, these methods fall short when there is *extreme* data heterogeneity among collaborating participants. We hypothesize that convergence under extreme data heterogeneity is primarily hindered due to the aggregation of conflicting updates from the participants in the initial collaboration rounds. To overcome this problem, we propose a warmup phase where each participant learns a personalized mask and updates only a subnetwork of the full model. This *personalized warmup* allows the participants to focus initially on learning specific *subnetworks* tailored to the heterogeneity of their data. After the warmup phase, the participants revert to standard federated optimization, where all parameters are communicated. We empirically demonstrate that the proposed personalized warmup via subnetworks (`FedPeWS`) approach improves accuracy and convergence speed over standard federated optimization methods.

## 1 Introduction

Federated learning (FL) is a distributed learning paradigm where participants collaboratively train a global model by performing local training on their data and periodically sharing local updates with the server. The server, in turn, aggregates the local updates to obtain the global model, which is then transmitted to the participants for the next round of training (McMahan et al., 2017). While FL preserves data confidentiality by avoiding collating participant data at the server, *statistical heterogeneity* between local data distributions is a significant challenge in FL (Kairouz et al., 2021). Several attempts have been made to tackle heterogeneity via federated optimization algorithms (Wang et al., 2019; Khaled et al., 2019; Li et al., 2020c;b; Karimireddy et al., 2020; Tupitsa et al., 2024; Sadiev et al., 2022; Beznosikov et al., 2021), dropout (Horvath et al., 2021; Alam et al., 2022), and batch normalization (Li et al., 2021d).

Consider the scenario where multiple hospitals collaborate to learn a medical image classification model that works across imaging modalities and organs, where the data from each hospital pertains to a different modality (e.g., histopathology, CT, X-ray, ultrasound, etc.) and/or organ (e.g., brain, kidney, colon, etc.). Most of the existing heterogeneous FL algorithms fail when there is such *extreme* data heterogeneity among collaborating participants, especially when the model is learned from scratch (with random initialization). The main reason for this failure is the high degree of conflicts between the local updates during the initial collaboration rounds. While enforcing a strong regularization constraint on the local updates (Li et al. (2020b)) can partially alleviate this problem, it dramatically slows down local learning and hence, convergence speed.

In this work, we explore an alternate approach to minimize the initial conflicts between heterogeneous participants by allowing participants in FL to initially train a partial subnetwork using only their local datasets. This warmup phase enables the participants to focus first on learning their local data well before engaging in broader collaboration. Thus, our proposed approach can be summarized as follows (see Figure 1). Initially, each participant uses a personalized binary mask tailored to their data distributions, allowing them to first learn their local data distributions and optimize their local (sparse) models. During this warmup phase, participants transmit only their masked updates to the

Figure 1: Conceptual illustration of training personalized subnetworks in federated learning.

server, and this process continues for a certain number of collaboration rounds. At the end of the warmup phase, the participants switch to standard federated optimization methods for subsequent collaboration. Our contributions are as follows:

1. We introduce a novel concept in federated learning, termed as *personalized warmup via subnetworks* (`FedPeWS`), which helps the global model to generalize to a better solution in fewer communication rounds. This is achieved through a neuron-level personalized masking strategy that is compatible with other FL optimization methods.

2. We propose an algorithm to *identify suitable subnetworks* (subset of neurons) for each participant by simultaneously learning the personalized masks and parameter updates. The proposed algorithm does not make any assumptions regarding the data distributions and incorporates a *mask diversity loss* to improve the coverage of all neurons in the global model.

3. For simple cases involving a small number of participants with known data distributions, we show that it is possible to skip the mask learning step and use fixed masks (that partition the network) determined by the server. We refer to this variant as `FedPeWS-Fixed`.

4. We empirically demonstrate the efficacy of the `FedPeWS` approach under both extreme non-i.i.d. and i.i.d. data scenarios using three datasets: a custom synthetic dataset, a combination of MNIST and CIFAR-10 datasets, and a combination of three distinct medical datasets (PathMNIST, OCTMNIST and TissueMNIST).

## 2    RELATED WORK

**Collaborative Learning.**    FL is a distributed learning paradigm that addresses data confidentiality concerns (Kairouz et al., 2021), particularly in environments where data can not be centralized due to regulatory or practical reasons (Albrecht, 2016). One of the seminal FL algorithms, FedAvg (McMahan et al., 2017), involves participants training models locally on their data and periodically transmitting their model parameters to a central server. The server averages these parameters to update the global model, which is then redistributed to the participants for further local refinement. FedAvg has inspired a plethora of variants and extensions aimed at enhancing performance (Karimireddy et al., 2020; Li et al., 2020b; Mishchenko et al., 2022), scalability (Guo et al., 2023; Al-Shedivat et al., 2021), communication efficiency (Ullah et al., 2023; Rahimi et al., 2023; Isik et al., 2023), privacy/confidentiality (Tastan & Nandakumar, 2023; Choquette-Choo et al., 2021; Ullah et al., 2023), robustness (Li et al., 2019), and fairness (Xu et al., 2021; Jiang et al., 2023; Tastan et al., 2024). For example, strategies such as weighted averaging or adaptive aggregation have been proposed to accommodate the non-i.i.d. nature of distributed data sources − a scenario where data is not identically distributed across all participants, which can significantly hinder model performance (Li et al., 2020b; Wang et al., 2020b; Karimireddy et al., 2020; Li et al., 2021d; Wang et al., 2020a). Specifically, FedProx (Li et al., 2020b) addresses data heterogeneity by integrating a proximal term into the FedAvg framework. There is also a body of work that focuses on addressing the heterogeneity problem through personalization-based approaches, utilizing local-centric objectives (Gasanov et al., 2022; Hanzely et al., 2023; Yoon et al., 2021; Li et al., 2021c).

**Independent Subnet Training.** Independent subnet training (IST) is a variant of distributed learning that focuses on enhancing model personalization and reducing communication overhead by training separate subnetworks for different participants (Yuan et al., 2022). IST distributes neurons of a fully connected neural network disjointly across different participants, forming a group of subnets. Then, each of these subnets is trained independently for one or more local SGD steps before synchronization. In every round, after broadcasting the server weights, each participant gets updated neurons to focus on, and the local subnet training continues. This approach led to different works along the line of using subnetwork training for efficiency (Horvath et al., 2021; Jiang et al., 2022; Diao et al., 2021; Nader et al., 2020; Alam et al., 2022; Li et al., 2021a; Mozaffari et al., 2021) in FL. In our work, we adopt IST's core principle of selecting neurons rather than focusing on weight values, which in turn narrows the search space. A key distinction between our method and IST lies in how the neurons are selected and the necessity of covering all neurons. Whilst IST typically involves random sampling of masks in each training round by the server and full coverage of neurons, we do not randomly sample neurons; instead, we use a learnable mask for each participant that is trained along with the parameters, and we relax the assumption of full coverage of neurons.

**Finding Subnetworks in FL.** Another relevant idea is the Lottery Ticket Hypothesis (LTH) (Frankle & Carbin, 2019), which attempts to identify subnetworks within a larger network. LTH is a model personalization technique, which focuses on sparsifying the network to create a smaller-scale version that improves per-round communication efficiency. In contrast to LTH, our method is directed towards training a shared global model and simultaneously improving convergence speed (reducing number of communication rounds). After LTH, there has been a growing interest in finding sparse and trainable networks at initialization (Mellor et al., 2021; Ji et al., 2021; Li et al., 2020a). Recently, in (Isik et al., 2023), sparse networks were found inside the main model to increase communication efficiency in FL. The proposed FedPM method focuses on finding a subnetwork by freezing the model weights and training for masks on a weight level, in contrast to IST, which works on a neuron level. FedPM utilizes the sigmoid function to obtain probability values from unbounded mask scores and then uses Bernoulli sampling to obtain binary masks. We use a similar approach in our `FedPeWS` algorithm to learn the neuron-level personalized masks.

## 3 PRELIMINARIES

Our goal is to minimize a sum-structured federated learning optimization objective:

$$x^\star \leftarrow \underset{x \in \mathbb{R}^d}{\arg\min} \left[ f(x) \coloneqq \frac{1}{N} \sum_{i=1}^{N} f_i(x) \right], \tag{1}$$

where the components $f_i : \mathbb{R}^d \to \mathbb{R}$ are distributed among $N$ local participants and are expressed in a stochastic format as $f_i(x) \coloneqq \mathbb{E}_{\xi \sim \mathcal{D}_i}\big[F_i(x, \xi)\big]$. Here, $\mathcal{D}_i$ represents the distribution of $\xi$ at participant $i \in [N] \coloneqq \{1, \ldots, N\}$. This problem encapsulates standard empirical risk minimization as a particular case when each $\mathcal{D}_i$ is represented by a finite set of $n_i$ elements, i.e., $\xi_i = \{\xi_i^1, \ldots, \xi_i^{n_i}\}$. In such cases, $f_i$ simplifies to $f_i(x, \xi_i) = \frac{1}{n_i} \sum_{j=1}^{n_i} F_i(x, \xi_i^j)$. Our approach does not impose restrictive assumptions on the data distribution $\mathcal{D}_i$. In fact, we specifically focus on the extreme heterogeneous (non-i.i.d.) setting, where $\mathcal{D}_i \neq \mathcal{D}_{i'}, \forall\, i \neq i'$ and the *local optimal solution* $x_i^\star \leftarrow \arg\min_{x \in \mathbb{R}^d} f_i(x)$ might significantly differ from the global minimizer of the objective function in Equation 1.

We are especially interested in the supervised classification task and let $\mathcal{M}_x : \mathcal{Z} \to \mathcal{Y}$ be a classifier parameterized by $x$. Here, $\mathcal{Z} \subseteq \mathbb{R}^D$ and $\mathcal{Y} = \{1, 2, \ldots, M\}$ denote the input and label spaces, respectively, $D$ is the input dimensionality, $M$ is the number of classes, and $d$ represents the number of parameters in the model $\mathcal{M}$. We set $F_i(x, \xi_i^j) = \mathcal{L}(\mathcal{M}_x(\mathbf{z}_i^j), y_i^j)$, where $\mathcal{L}$ is an appropriate loss function and $\xi_i^j \coloneqq (\mathbf{z}_i^j, y_i^j)$ is a labeled training sample such that $\mathbf{z}_i^j \in \mathcal{Z}$ and $y_i^j \in \mathcal{Y}$. Furthermore, we mainly focus on the cross-silo FL setting ($N$ is small).

**Federated Averaging (FedAvg).** A common approach for solving Equation 1 in the distributed setting is FedAvg (McMahan et al., 2017). This algorithm involves the participants performing $K$ local steps of stochastic gradient descent (SGD) and communicating with the server over $T$ communication rounds. The server initializes the global model with $x_g^0$ and broadcasts it to all

---

**Algorithm 1** `FedPeWS` (For `FedPeWS-Fixed` variant, the steps highlighted in green are omitted and instead the server sets $m_i^t = m_i, \forall\, t \in [W]$.)

---

**Input:** Number of collaboration rounds $T$, number of warmup rounds $W$, number of local steps $K$, local learning rate $\eta_\ell$, global learning rate $\eta_g$, mask learning rate $\eta_s$, $\lambda$ (mask diversity weight)

1: Initialize $x_g^0$ and $s_g^0$, compute $\theta_g^0 = \sigma(s_g^0)$
2: **for** $t = 1, \dots, T$ **do**
3:      **if** $t > W$ **then** // Use all parameters after warmup
4:         Set $m_i^t = \mathbf{1}$, i.e., $m_i(\ell) = 1, \forall\, \ell \in [d]$
5:      **end if**
6:      Server sends global model $x_g^{t-1}$ and global mask probability $\theta_g^{t-1}$ to all clients $i \in [N]$
7:      **for** client $i \in [N]$ in parallel **do**
8:         Initialize local model $x_i^{t,0} \leftarrow x_g^{t-1}$
9:         $s_i^{t,0} \leftarrow s_g^0$ **if** $t = 1$ **else** $s_i^{t,0} \leftarrow s_i^{t-1,K}$ **endif**
10:         **for** $k = 1, \dots, K$ **do**
11:            **Procedure I**: Freeze model weights $x_i^{t,k-1}$
12:            Optimize over $s$: $\mathcal{L}_s = f_i\left(x_i^{t,k-1} \odot \mathcal{G}\left(s_i^{t,k-1}\right), \xi_i^{t,k-1}\right) - \lambda \|\sigma\left(s_i^{t,k-1}\right) - \theta_{g\setminus\{i\}}^{t-1}\|_2^2$
13:            Update: $s_i^{t,k} \leftarrow s_i^{t,k-1} - \eta_s \nabla_s \mathcal{L}_s$
14:            **Procedure II.** Freeze mask score vector $s_i^{t,k}$
15:            Optimize over $x$ : $\mathcal{L}_x = f_i\left(x_i^{t,k-1} \odot \mathcal{G}\left(s_i^{t,k}\right), \xi_i^{t,k-1}\right)$
16:            Update: $x_i^{t,k} \leftarrow x_i^{t,k-1} - \eta_\ell \nabla_x \mathcal{L}_x$
17:         **end for**
18:         Compute $m_i^t = \mathcal{G}(s_i^{t,K})$ and upload $x_i^t \leftarrow x_i^{t,K}$, $m_i^t$ to server
19:      **end for**
20:      $x_g^t = x_g^{t-1} - \eta_g \left( x_g^{t-1} - \dfrac{\sum_{i \in [N]} x_i^t \odot m_i^t}{\sum_{i \in [N]} m_i^t} \right)$
21: **end for**

---

participants, which is then used to initialize the local models, i.e., $x_i^{1,0} = x_g^0$. In each communication round, the updates from the participants are averaged on the server and sent back to all participants. For a local step $k \in [K]$, communication round $t \in [T]$, and participant $i \in [N]$, the local and global iterates are updated as:

$$x_i^{t,k} = x_i^{t,k-1} - \eta_\ell \nabla f_i\big(x_i^{t,k-1}, \xi_i^{t,k-1}\big), \quad x_i^t = x_i^{t,K}, \text{ and } x_g^t = x_g^{t-1} - \eta_g\big(x_g^{t-1} - \frac{1}{N}\sum_{i=1}^{N} x_i^t\big), \tag{2}$$

where $\eta_\ell$ and $\eta_g$ are the local and global learning rates, respectively. The server then broadcasts the updated global model $x_g^t$ to all participants, which is then used to reinitialize the local models as $x_i^{t+1,0} = x_g^t$.

In the FedAvg algorithm, the number of communication rounds necessary to achieve a certain precision is directly proportional to the heterogeneity measure (Li et al., 2020c). Notably, this relationship holds true in convex settings; however, in non-convex scenarios, the algorithm may either not converge or may converge to a suboptimal solution. Stemming from this observation, our objective is to reduce the number of requisite communication rounds to achieve convergence, while simultaneously achieving a better solution.

## 4 PROPOSED FEDPEWS METHOD

The core idea of the proposed `FedPeWS` method is to allow participants to learn only a personalized subnetwork (a subset of parameters) instead of the entire network (all parameters) during the initial

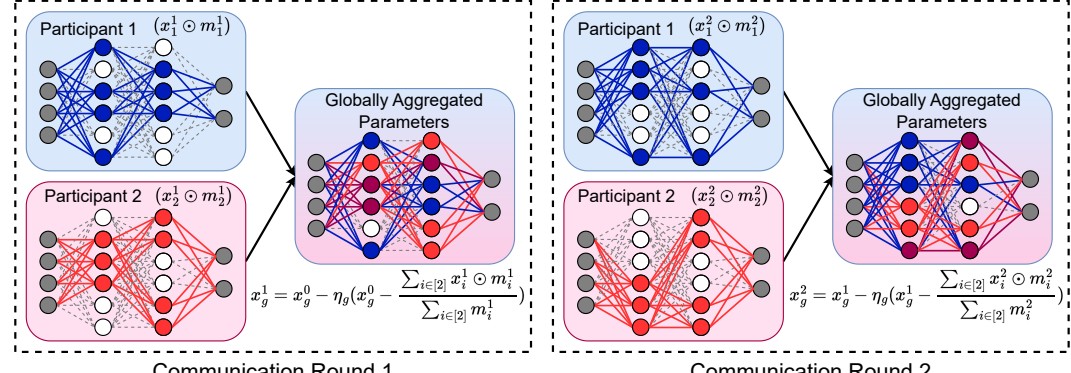

Figure 2: Illustration of the proposed FedPeWS algorithm for two participants, which aggregates partial subnetworks $(x_i^t \odot m_i^t)$ during the warmup phase to obtain a shared global model $x_g^t$. Here, $x_i^t$ and $m_i^t$ denote the local model and personalized mask of the $i^{\text{th}}$ participant in the $t^{\text{th}}$ round.

warmup phase. Let $m_i \in \{0,1\}^d$ be a binary mask vector denoting the set of parameters that are learned by participant $i$, $i \in [N]$. Note that $m_i(\ell) = 1$ indicates that the $\ell^{\text{th}}$ element of $x_i$ is selected for learning (value 0 indicates non-selection), $\ell \in [d]$. Thus, during the warmup phase, the objective in FedPeWS is to learn the parameters $x$ along with the personalized masks $m_i$, i.e.,

$$\min_{x, \{m_i\}_{i \in [N]}} \frac{1}{N} \sum_{i=1}^{N} f_i(x \odot m_i), \qquad (3)$$

$\odot$ denotes element-wise multiplication. Note that $\mathcal{M}_{x \odot m_i}$ denotes the personalized subnetwork of participant $i$. When personalized masks are employed, the update rules can be modified as:

$$x_i^{t,k} = x_i^{t,k-1} - \eta_\ell \nabla f_i\big(x_i^{t,k-1} \odot m_i^t, \xi_i^{t,k-1}\big) \text{ and } x_g^t = x_g^{t-1} - \eta_g\big(x_g^{t-1} - \frac{\sum_{i \in [N]} x_i^t \odot m_i^t}{\sum_{i \in [N]} m_i^t}\big). \quad (4)$$

The obvious questions regarding the FedPeWS method are: (i) how to learn these personalized masks $m_i$? and (ii) what should be the length of the warmup period?

**Identification of personalized subnetworks**: It is not straightforward to directly optimize for the personalized binary (discrete) masks $m_i$ in Equation 3. Hence, we make the following design choices. Firstly, personalized masks are learned at the neuron-level and then expanded to the parameter-level. Following IST (Yuan et al., 2022), masks are specifically applied only to the hidden layer neurons, while the head and tail neurons remain unaffected. However, unlike IST, the neuron-level masks are not randomly selected in each collaboration round. Instead, we learn real-valued personalized neuron-level mask score vectors $s_i \in \mathbb{R}^h$, which in turn can be used to generate the binary masks. Here, $h$ denotes the number of hidden neurons in the classifier $\mathcal{M}$ and $h \ll d$. A higher value of element $s_i(\ell)$, $\ell \in [h]$, indicates that the $\ell^{\text{th}}$ neuron is more likely to be selected by participant $i$. Let $\mathcal{G} : \mathbb{R}^h \to \{0,1\}^d$ be the mask generation function that generates the binary parameter-level masks $m_i$ from neuron-level mask score vectors $s_i$, i.e., $m_i = \mathcal{G}(s_i)$. $\mathcal{G}$ consists of three steps. Firstly, we convert $s_i$ into probabilities by applying a sigmoid function, i.e., $\theta_i = \sigma(s_i)$, where $\theta_i \in [0,1]^h$ is the mask probability vector and $\sigma$ is the sigmoid function. Next, binary neuron masks $\tilde{m}_i$ are obtained by sampling from a Bernoulli distribution with parameter $\theta_i$, i.e., $\tilde{m}_i(\ell) \sim Bernoulli(\theta_i(\ell))$, $\forall \ell \in [h]$. Finally, these binary neuron masks can be directly mapped to the binary parameter-level mask $m_i$, i.e., if a neuron is selected, all the weights associated with the selected neuron are also selected. Thus, Equation 3 can be reparameterized as:

$$\min_{x, \{s_i\}_{i \in [N]}} \frac{1}{N} \sum_{i=1}^{N} f_i(x \odot \mathcal{G}(s_i)). \qquad (5)$$

The above equation can be optimized alternatively for the mask score vectors $s_i$ and the parameters $x$. The participants first optimize for the mask scores while the model parameters $x_i^{t,k}$ are frozen

(Procedure I), and then switch to optimizing the model parameters while freezing the mask scores (Procedure II). In the mask training step (Procedure I), the optimization objective is defined as:

$$\mathcal{L}_s = f_i\left(x_i^{t,k} \odot \mathcal{G}\left(s_i^{t,k}\right), \xi_i^{t,k-1}\right) - \lambda \|\sigma\left(s_i^{t,k}\right) - \theta_{g\setminus\{i\}}^t\|_2^2; \qquad s_i^{t,k+1} \leftarrow s_i^{t,k} - \eta_s \nabla_s \mathcal{L}_s, \quad (6)$$

where $\nabla_s$ indicates that the gradient is w.r.t. mask score vector $s$, $\eta_s$ is the local learning rate for updating $s$, $\theta_g^t$ is the global mask probability at round $t$, $\theta_{g\setminus\{i\}}^t$ is the global mask probability excluding the probability mask of the current participant $i$, and $\lambda$ is the weight assigned to the mask diversity measure (second term). It is important to note that the personalized masks may not cover all neurons in the network. Maximizing the mask diversity measure encourages personalized masks to deviate as much as possible from the global mask, which facilitates better coverage of all the neurons in the global model. The diversity measure has an upper bound due to the sigmoid function:

$$\|\sigma\left(s_i^{t,k}\right) - \theta_{g\setminus\{i\}}^t\|_2^2 \leq h. \qquad (7)$$

Given the difficulty in calculating $\nabla_s \mathcal{L}_s$ directly due to the discrete nature of Bernoulli sampling, we employ the straight-through estimator (STE) (Bengio et al., 2013) to approximate the gradients, which does not compute the gradient of the given function and passes on the incoming gradient as if the function was an identity function.

During Procedure II, the optimization function for the model weights is expressed as:

$$\mathcal{L}_x = f_i\left(x_i^{t,k} \odot \mathcal{G}\left(s_i^{t,k}\right), \xi_i^{t,k-1}\right); \qquad x_i^{t,k+1} \leftarrow x_i^{t,k} - \eta_\ell \nabla_x \mathcal{L}_x, \qquad (8)$$

where $\nabla_x$ indicates that the gradient is w.r.t. weights $x$. The `FedPeWS` algorithm alternates between these two procedures for $W$ rounds, where $W$ is the number of warmup rounds. At this point, the warmup stops and the participants switch to standard training for $(T - W)$ collaboration rounds. This approach ensures that each participant effectively contributes to the FL process while also tailoring the learning to their specific data distributions. The number of warmup rounds $W$ (or the proportion of warmup rounds $\tau = \frac{W}{T}$) is a key hyperparameter of the `FedPeWS` algorithm, along with the weight $\lambda$ assigned to the mask diversity loss. While it would be ideal to have a principled method to select these hyperparameters, we use a grid search to tune them, which is currently a limitation.

**Use of fixed subnetworks**: When the number of participants is small and the data distributions of the participants are known apriori, the server can partition the full model into subnetworks of the same depth and assign a fixed subnetwork to each participant, i.e., $m_i^t = m_i, \forall\, t \in [W]$. Participants transmit only the masked updates back to the server during warmup, which then aggregates these masked parameters and redistributes them in their masked form. For the sake of utility, the server can design personalized masks such that the union of these masks covers all the neurons. This variant of `FedPeWS` is referred to as `FedPeWS-Fixed` and follows the same algorithm in Algorithm 1, except for the omission of the highlighted (green) steps.

## 5 EXPERIMENTS AND RESULTS

### 5.1 DATASETS AND NETWORK ARCHITECTURE

**Synthetic Dataset.** To effectively evaluate the performance of the proposed algorithm, we generated a custom synthetic dataset that simulates the extreme non i.i.d. scenario. This dataset encompasses four classes, each characterized by four 2D clusters determined by specific centers and covariance matrices. Note that the clusters from different classes interleave each other as shown in Figure 3. For this dataset, we utilize a neural network consisting of five fully-connected (FC) layers, each followed by ReLU activation functions, except the last layer. To enhance the dataset complexity and aid FC network learning, we transform these 2D points into 5D space using the transformation $[x, y, x^2, y^2, xy]$, based on their $(x, y)$ coordinates. We generate two versions of this dataset, **Synthetic-32K** and **Synthetic-3.2K**, depending on the number of data points in the training set. The former has 32000 samples, with each class containing 8000 data points, while the latter has ten times fewer data points.

**CIFAR-MNIST.** We integrate two distinct datasets, CIFAR-10 (Krizhevsky et al., 2009) and MNIST (LeCun, 1998), to explore how different clients might adapt when faced with disparate data sources. CIFAR-10 comprises of $32 \times 32$ pixel images categorized into 10 object classes. MNIST, typically featuring $28 \times 28$ pixel images across 10 digit classes, is upscaled to $32 \times 32$ pixel to standardize dimensions with CIFAR-10. We compile a balanced dataset by randomly selecting 400 samples from each class for the training set and 200 samples for the test set

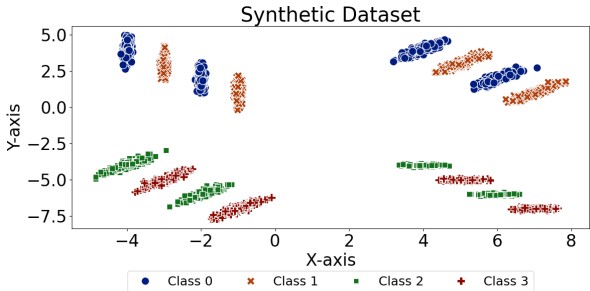

Figure 3: Samples from the custom synthetic dataset.

from the combined pool of 20 classes. This setup aims to simulate a FL environment where multiple clients handle significantly varied data types. For this dataset, we employ a convolutional neural network comprising four convolutional layers, each having a kernel size of 3 and padding of 1, followed by max pooling. This is succeeded by a fully connected layer. This architecture was used because of its simplicity and widespread use in the literature (Yuan et al., 2022; Isik et al., 2023).

**{Path-OCT-Tissue}MNIST.** We amalgamate three distinct medical datasets: PathMNIST, OCTM-NIST, and TissueMNIST (Yang et al., 2023), to develop a universal medical prognosis model capable of recognizing various tasks using a single model. The datasets contain 9, 4, and 8 classes, respectively, totaling to 21 classes. For this dataset, we utilized the same architecture and training details described in the CIFAR-MNIST dataset.

## 5.2 EXPERIMENTAL SETUP

**Dataset partitioning.** For scenarios with a smaller number of collaborators ($N = 2, 3, 4$), we manually partition the training dataset to tailor the data distribution to specific participants. In the $N = 2$ scenario, we partition as follows: (i) For the Synthetic dataset, encompassing both Synthetic-32K and Synthetic-3.2K, even-numbered classes are assigned to participant 1, while odd-numbered classes are allocated to participant 2. (ii) For the CIFAR-MNIST combination, all CIFAR-10 samples are assigned to participant 1, with MNIST samples allocated to participant 2. In the $N = 3$ scenario, the {Path-OCT-Tissue}MNIST dataset is partitioned into three splits corresponding to the individual datasets, with PathMNIST assigned to participant 1, OCTMNIST to participant 2, and TissueMNIST to participant 3. For the $N = 4$ scenario, the synthetic dataset is divided so that each class is exclusively allocated to one of the four participants.

For scenarios with a larger number of participants ($N \geq 10$), we employ a Dirichlet distribution to partition the training set. This approach utilizes a concentration parameter $\alpha$ to simulate both homogeneous and heterogeneous data distributions (Yurochkin et al., 2019; Li et al., 2021b; Lin et al., 2020; Wang et al., 2020a). We experiment with various values of $\alpha$, specifically $\alpha \in \{0.1, 0.5, 1.0, 2.0, 5.0\}$, to explore the effects of dataset heterogeneity (lower $\alpha$ values) and homogeneity (higher $\alpha$ values) on the model performance. This methodological diversity allows us to comprehensively assess our approach under varying data conditions. Results for large $N$ ($> 100$) are reported in the appendix.

**Training details.** In federated optimization, we primarily benchmark against the FedAvg algorithm (McMahan et al., 2017), a standard approach in federated learning. However, our algorithm is designed to be versatile, functioning as a 'plug-and-play' solution that is compatible with various other optimizers. To demonstrate this adaptability, we also conduct experiments using FedProx (Li et al., 2020b), showcasing our method's capabilities across different optimization frameworks. For our experiments, we fix the local learning rate $\eta_\ell = 0.001$ in the Synthetic-32K dataset case, and we set $\eta_\ell = 0.01$ for other experiments. Also, the mask learning rate is fixed $\eta_s = 0.1$. Furthermore, we vary the global learning rate $\eta_g \in \{0.1, 0.25, 0.5, 1.0\}$ to observe the differences in optimization behavior between the baseline and our proposed methods. Additionally, we employ two distinct batch sizes $\{32, 8\}$ for Synthetic-32K and Synthetic-3.2K, respectively. For experiments involving the CIFAR-MNIST and {Path-OCT-Tissue}MNIST datasets, we standardize the batch size to 64. We conduct our experiments on NVIDIA RTX A6000 GPUs on an internal cluster server, with each

Table 1: The required number of collaboration rounds to reach target accuracy $\upsilon$ % and the final accuracy after $T$ rounds. The results are averaged over 3 seeds. $\times$ indicates that the algorithm cannot reach target accuracy $\upsilon$ within $T$ rounds and NA means that it reaches $\upsilon$ only in one random seed.

| Dataset / Batch size | | Synthetic-32K, 32 | | | Synthetic-3.2K, 8 |
|---|---|---|---|---|---|
| Parameters $\{\eta_g/\lambda/\tau\}$ | | $\{1.0/5.0/0.125\}$ | $\{0.5/2.0/0.2\}$ | $\{0.25/1.0/0.1875\}$ | $\{0.1/2.0/0.1\}$ |
| Target accuracy $\upsilon(\%)$ | | 99 | 90 | 75 | 99 |
| No. of rounds to reach target accuracy | FedAvg | $148 \pm 3.79$ | $199 \pm$ NA | $\times$ | $371 \pm$ NA |
| | FedPeWS | $\mathbf{115 \pm 7.21}$ | $\mathbf{182 \pm 6.81}$ | $\mathbf{286 \pm 7.93}$ | $\mathbf{301 \pm 10.59}$ |
| Final accuracy after $T$ collaboration rounds | FedAvg | $99.94 \pm 0.05$ | $91.40 \pm 7.25$ | $67.64 \pm 0.90$ | $97.33 \pm 3.89$ |
| | FedPeWS | $\mathbf{99.96 \pm 0.01}$ | $\mathbf{99.49 \pm 0.60}$ | $\mathbf{83.50 \pm 3.52}$ | $\mathbf{99.66 \pm 0.19}$ |

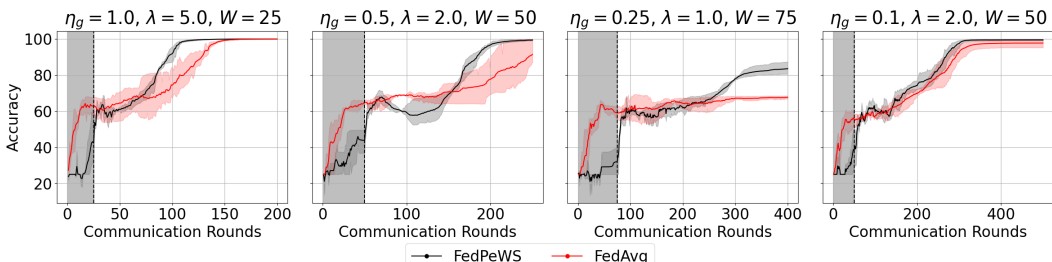

Figure 4: Results of the experiments on Synthetic-$\{32, 3.2\}$K datasets with batch sizes $\{32, 8\}$, with different global learning rates $\eta_g \in \{1.0, 0.5, 0.25, 0.1\}$ and communication rounds $T \in \{200, 250, 400, 500\}$. Refer to Table 1 for the corresponding numbers. In all the above scenarios, FedPeWS converges faster to a better solution compared to FedAvg.

run utilizing a single GPU. The execution time for each run is capped at less than an hour, which indicates the maximum execution time rather than the average. All results are averaged over three independent runs and the average accuracy is reported on the global test dataset.

## 5.3 EXPERIMENTAL RESULTS

Our experimental analysis focuses on assessing the performance of our proposed FedPeWS algorithm within the FL framework. The key findings from our studies are as follows: (i) The FedPeWS approach demonstrates a significant reduction in the number of communication rounds required to achieve target accuracy while also enhancing the final accuracy post-convergence. (ii) The FedPeWS algorithm is robust across different levels of data heterogeneity. (iii) In scenarios where full knowledge of the participant data distributions is available, the server can employ the FedPeWS-Fixed

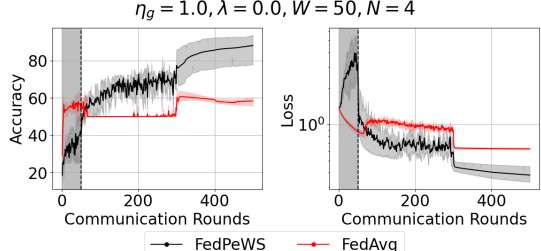

Figure 5: Visualization of validation accuracy and loss on the Synthetic-32K dataset with $N = 4$ participants and a global learning rate $\eta_g = 1.0$.

method (Figure 6). While the FedPeWS-Fixed variant shows competitive effectiveness comparable to our primary FedPeWS algorithm, the latter offers broader applicability in real-world settings.

**Improved communication efficiency and accuracy.** We initially report the required number of communication rounds to reach the target accuracy and the final accuracy after $T$ communication rounds for the synthetic dataset in Table 1. The results underscore that the incorporation of a personalized warmup phase in a federated setup significantly reduces the required number of communication rounds across all tested scenarios. Notably, in specific instances, such as with the Synthetic-32K dataset and $\eta_g = 0.25$, the conventional FedAvg algorithm does not meet the target accuracy within the $T$ communication rounds. Conversely, in scenarios where $\eta_g \in \{0.25, 0.1\}$,

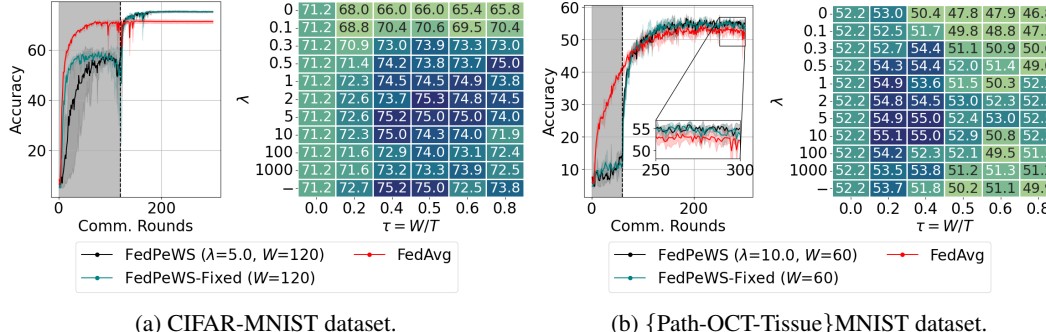

(a) CIFAR-MNIST dataset.

(b) {Path-OCT-Tissue}MNIST dataset.

Figure 6: Results for experiments on (a) the CIFAR-MNIST and (b) {Path-OCT-Tissue}MNIST datasets with a communication budget of $T = 300$. **(a) Left:** Participant 1 has MNIST data samples; Participant 2 has CIFAR-10 data samples. **(a) Right:** Ablation study for $\lambda$ and $\tau$ parameters on CIFAR-MNIST (see Table 6). **(b) Left:** Each of $N = 3$ participants holds unique dataset samples from {PathMNIST, OCTMNIST, TissueMNIST} pool. **(b) Right:** Ablation study for $\lambda$ and $\tau$ on the respective dataset (see Table 7). The first column ($\tau = 0.0$) corresponds to the FedAvg algorithm. The last row presents results for the `FedPeWS-Fixed` algorithm.

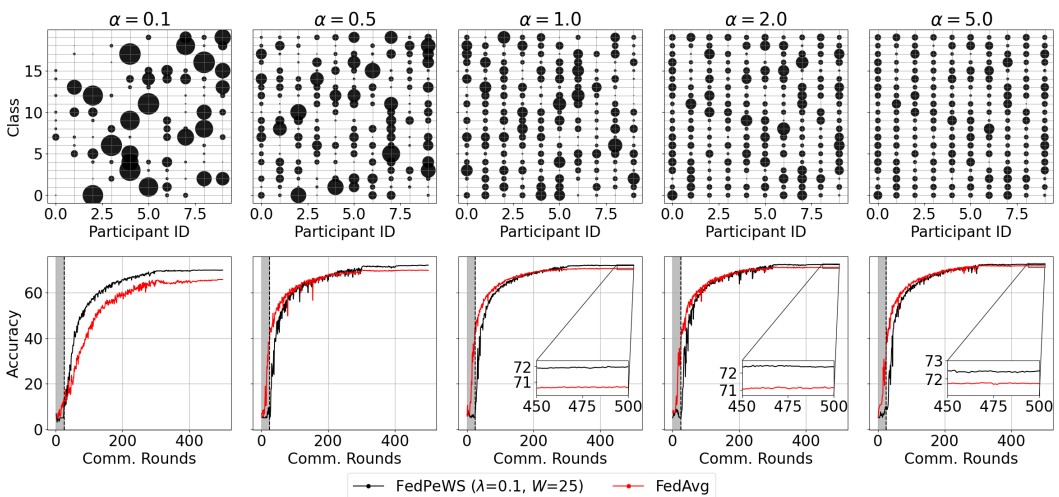

Figure 7: Top: illustration of number of samples per class allocated to each client, that is indicated by dot sizes, for different concentration $\alpha$ values. Bottom: visualization of the experiments on CIFAR-MNIST dataset with $N = 10$ participants with different levels of heterogeneity.

FedAvg only achieves the target accuracy in one of the seeds, exhibiting suboptimal performance in the other two runs. From Figure 4, it is evident that our proposed `FedPeWS` algorithm surpasses FedAvg in both communication efficiency and accuracy.

We also consider a more extreme data heterogeneity scenario with $N = 4$ participants, depicted in Figure 5, where FedAvg completely fails by reaching only $58.4 \pm 2.33\%$, whereas our `FedPeWS` approach reaches $91.13 \pm 3.55\%$ accuracy by significantly outperforming the base optimizer (FedAvg) with a gain of **32.72%**. It is crucial to highlight that in this experiment, we set $\lambda = 0.0$, effectively not enforcing diversity as outlined in Equation 6. This approach focuses solely on optimizing the masks using the first loss component, which depends only on the data distributions of each participant. This shows that, in specific scenarios, we can learn the personalized masks (Procedure I) without the need to adjust the $\lambda$ parameter, while still achieving a better performance than the base optimizer.

**Sensitivity to $\lambda$ and $\tau$ parameters.** Figure 6 showcases the results of experiments on the CIFAR-MNIST dataset with $N = 2$ participants and {Path-OCT-Tissue}MNIST dataset with $N = 3$ participants. The left-side plots of Figures 6a and 6b, which depict the performance of the global

Table 2: The required number of collaboration rounds to reach target accuracy $\upsilon$ % using FedProx algorithm and the final accuracy after $T$ rounds. The results are averaged over 3 seeds. $\times$ indicates that the algorithm cannot reach target accuracy $\upsilon$ within $T$ rounds.

| Dataset / Batch size | | Synthetic-32K, 32 | | | Synthetic-3.2K, 8 |
|---|---|---|---|---|---|
| Parameters $\{\eta_g/\lambda/\tau\}$ | | $\{1.0/0.1/0.125\}$ | $\{0.5/0.1/0.2\}$ | $\{0.25/1.0/0.1875\}$ | $\{0.1/1.0/0.1\}$ |
| Target accuracy $\upsilon(\%)$ | | 99 | 90 | 75 | 99 |
| No. of rounds to reach target accuracy | FedProx | $138 \pm 13.22$ | $\times$ | $\times$ | $362 \pm 20.00$ |
| | FedPeWS | $\mathbf{115 \pm 5.29}$ | $\mathbf{211 \pm 16.52}$ | $\mathbf{314 \pm 27.83}$ | $\mathbf{344 \pm 27.30}$ |
| Final accuracy after $T$ collaboration rounds | FedProx | $99.95 \pm 0.02$ | $82.43 \pm 7.98$ | $69.26 \pm 6.03$ | $\mathbf{99.92 \pm 0.06}$ |
| | FedPeWS | $\mathbf{99.96 \pm 0.01}$ | $\mathbf{98.40 \pm 1.84}$ | $\mathbf{90.40 \pm 3.91}$ | $\mathbf{99.92 \pm 0.07}$ |

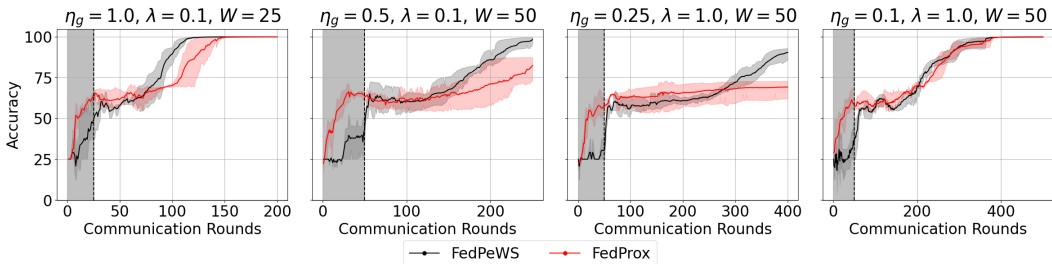

Figure 8: Comparison of our proposed method and FedProx (Li et al., 2020b) on Synthetic-{32, 3.2}K datasets. Refer to Table 2 for the corresponding numbers.

model (averaged over 3 runs), demonstrate that our method consistently achieves higher accuracy. The right side figures feature heatmap plots that annotate the global model accuracy obtained varying $\lambda \in \{0, 0.1, 0.3, 0.5, 1, 2, 5, 10, 100, 1000\}$ and $\tau \in \{0.0, 0.2, 0.4, 0.5, 0.6, 0.8\}$ parameters. An additional row labeled $(\lambda = -)$ represents the FedPeWS-Fixed approach, where user(server)-defined fixed masks are employed. In this method, we simply split the full network into $N$ partitions, with each partition assigned to a participant (for detailed instructions on setting masks, please see Section B.2 in the appendix). The results indicate that our approach has a low sensitivity to variations in $\lambda$ and $\tau$. For more detailed insights, please refer to Tables 6 and 7 in the appendix.

**Varying degrees of heterogeneity.** Figure 7 demonstrates that our FedPeWS approach consistently outperforms FedAvg, with gains directly related to the degree of data heterogeneity. The figure clearly shows that the advantage of using our method is more pronounced under conditions of high data heterogeneity. As heterogeneity levels decrease, our method becomes comparable to FedAvg.

**FedProx.** We also present results using the FedProx optimizer on Synthetic-32K and Synthetic-3.2K datasets in Figure 8 and Table 2, employing global learning rates $\eta_g = \{1.0, 0.5, 0.25, 0.1\}$. Note that we adapt Algorithm 1 to incorporate the FedProx algorithm as the base optimizer, instead of FedAvg. We selected the best performing proximal term scaler 0.01 after tuning and evaluating different values from a set of potential values $\{0.001, 0.01, 0.1, 0.5\}$, based on the findings in (Li et al., 2020b). The results demonstrate that FedPeWS outperforms FedProx in terms of both communication efficiency and final accuracy across the tested scenarios, except the last scenario (Synthetic-3.2K dataset with batch size 8 and $\eta_g = 0.1$), where the performance of FedPeWS is comparable to that of FedProx.

# 6 CONCLUSION

In this work, we introduced a novel concept called *personalized warmup via subnetworks* for heterogeneous FL − a strategy that enhances convergence speed and can seamlessly integrate with existing optimization techniques. Results demonstrate that the proposed FedPeWS approach and achieves higher accuracy than the relevant baselines, especially when there is extreme statistical heterogeneity. Limitations of FedPeWS include the need to tune two additional hyperparameters (no. of warmup rounds and mask diversity weight) and the lack of theoretical convergence analysis.

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

## A    RELATED WORK

Table 3 shows the comparison of our proposed approach to the existing literature.

Table 3: Comparison of approaches for handling data heterogeneity in federated learning.

| | Shared Global Model | Level of Mask Personalization | Learnable Mask | Learnable Parameters |
|---|---|---|---|---|
| FedPM (Isik et al., 2023) | ✓ | parameter-level | ✓ | ✗ |
| IST (Yuan et al., 2022) | ✓ | neuron-level | ✗ (random) | ✓ |
| LTH (Frankle & Carbin, 2019) | ✗ | parameter-level | ✗ (pruned) | ✓ |
| FedWeIT (Yoon et al., 2021) | ✗ | parameter-level | ✓ | ✓ |
| FjORD (Horvath et al., 2021) | ✓ | parameter-level | ✗ (slimmed) | ✓ |
| FedPeWS (ours) | ✓ | neuron-level | ✓ | ✓ |

## B    ADDITIONAL EXPERIMENTAL DETAILS

### B.1    NETWORK ARCHITECTURE DETAILS

The network for the synthetic dataset (detailed in Section 5.2) consists of five fully connected (FC) layers, each followed by ReLU activation functions, except for the last layer. We provide the details of this architecture in Table 4.

Table 4: Architecture for synthetic dataset models used in the experiments.

| Layer | Input | FC1 | FC2 | FC3 | FC4 | FC5 |
|---|---|---|---|---|---|---|
| Dimensions | [5] | [5, 32] | [32, 64] | [64, 128] | [128, 32] | [32, 4] |

The network for the CIFAR-MNIST and {Path-OCT-Tissue}MNIST datasets includes three convolutional layers followed by max pooling, and a fully connected layer. The details of this architecture is provided in Table 5.

Table 5: Architecture for CIFAR-MNIST dataset models. Every convolutional layer is followed by a max pooling layer with kernel size 2 and stride 2.

| Layer | Input | Conv1 | Conv2 | Conv3 | Flatten | FC |
|---|---|---|---|---|---|---|
| Dimensions | [3, 32, 32] | [3, 32, 3, 3] | [32, 64, 3, 3] | [64, 128, 3, 3] | [2048] | [2048, 20] |

### B.2    FIXED MASK GENERATION

Figure 9 illustrates how we design masks for FedPeWS-Fixed experiments in scenarios with $N = 2$ participants. For cases involving $N = 4$ participants, the full network $\mathcal{M}_x$ (classifier $\mathcal{M}$ parameterized with $x$) is divided into four subnetworks, vertically, with each subnetwork corresponding to one of the participants. As such, we vertically partition the hidden neurons in the network into $N$ groups (subnetworks) and design the mask to assign each group to one participant, ensuring no overlap. This design choice is based on the assumption that classes held by each participant are highly heterogeneous, thus preventing any intersection in the masks. This setting is specifically tailored for the FedPeWS-Fixed method and doesn't necessitate performing optimization over the masks $m_i$, they are kept fixed.

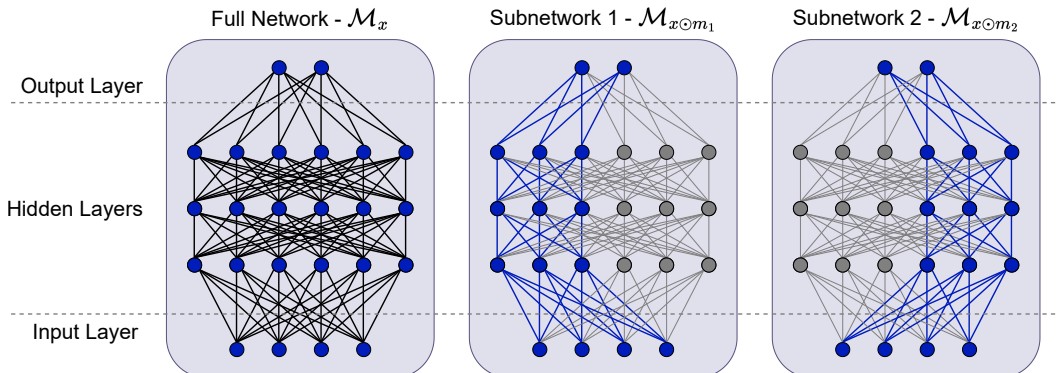

Figure 9: Illustration of manual mask setting in the `FedPeWS-Fixed` method. The left figure illustrates the complete network with all neurons active and full connections. The middle figure represents subnetwork 1, utilizing only the left portion of the full network, where $m_1$ corresponds to this left side. Conversely, the right figure indicates the part of the network used for subnetwork 2. This setting is employed in all experiments involving $N = 2$ participants.

## C EXPERIMENTAL RESULTS

### C.1 WALL-CLOCK TIME VS. ACCURACY

Figure 10 illustrates the wall-clock time versus accuracy results, which correspond to Figure 4 in the main paper. From this comparison, `FedPeWS` demonstrates a slightly improved performance over FedAvg in terms of wall-clock time in two of the four scenarios. However, it underperforms slightly in the remaining two scenarios, with only a marginal increase in time. This variance is attributed to the alternation between training masks and weights during the warm-up phase, impacting the time efficiency.

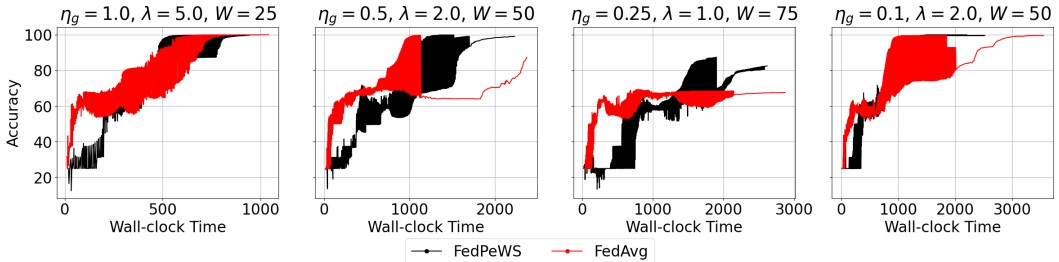

Figure 10: Wall-clock time vs. accuracy plot corresponding to Figure 4 of the main paper.

### C.2 FEDPEWS–FIXED. MASK LENGTH STUDY

In this section, we explore the impact of mask length on the performance of the `FedPeWS-Fixed` method with parameter $W = 120(\tau = 0.4)$ using the CIFAR-MNIST dataset. We examine two scenarios for splitting the network into two subnetworks:

1. $|m_1| < |m_2|$: 75% of the mask is assigned to Participant 2, 25% to Participant 1.
2. $|m_1| = |m_2|$: equal sized masks are assigned to each participant.

Figure 11 displays the validation accuracy over $T = 300$ communication rounds for both scenarios. The leftmost plot shows the accuracy of the global model, while the middle and rightmost plots the accuracy for each of the participants. Both mask length scenarios converge to a comparable accuracy levels, with a marginal difference of $0.5\%$ higher accuracy in the scenario where $|m_1| < |m_2|$. This is likely due to the larger mask size, which aids in learning the more complex CIFAR-10 dataset held by Participant 2.

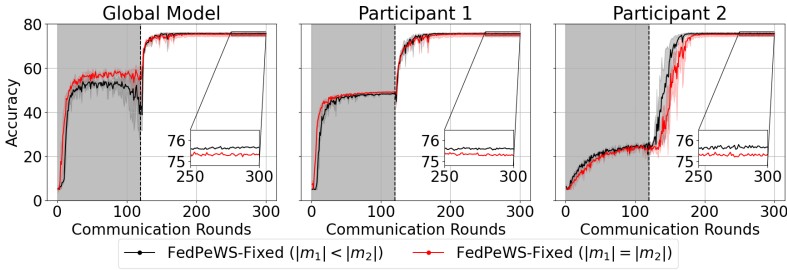

Figure 11: Mask length study using `FedPeWS-Fixed` method on CIFAR-MNIST dataset.

## C.3 SENSITIVITY ANALYSIS

In this section, we detail the sensitivity analysis of the $\lambda$ and $\tau$ parameters conducted on the CIFAR-MNIST dataset (Table 6) and the {Path-OCT-Tissue}MNIST dataset (Table 7). This analysis particularly includes the standard deviation of the accuracy achieved by the tested algorithms after $T$ communication rounds and over three independent evaluations. Results with the best performance are highlighted in green.

Table 6: Ablation study of the parameters $\lambda$ and $\tau$ on the CIFAR-MNIST dataset with $N = 2$ participants over three independent runs. The first column ($\tau = 0.0$) corresponds to the FedAvg algorithm. The last row $(-)$ presents results for the `FedPeWS-Fixed` algorithm.

| $\lambda(\downarrow), \tau(\rightarrow)$ | $\tau = 0.0$ (FedAvg) | Proportion of warmup rounds $\tau = W/T$ | | | | |
|---|---|---|---|---|---|---|
| | | $\tau = 0.2$ | $\tau = 0.4$ | $\tau = 0.5$ | $\tau = 0.6$ | $\tau = 0.8$ |
| 0.0 | | $68.01 \pm 0.88$ | $66.01 \pm 0.48$ | $65.96 \pm 1.03$ | $65.40 \pm 1.95$ | $65.77 \pm 0.41$ |
| 0.1 | | $68.77 \pm 1.09$ | $70.39 \pm 1.00$ | $70.61 \pm 1.17$ | $69.48 \pm 0.37$ | $70.36 \pm 2.06$ |
| 0.3 | | $70.86 \pm 0.23$ | $73.00 \pm 0.65$ | $73.91 \pm 0.71$ | $73.26 \pm 0.46$ | $73.02 \pm 1.05$ |
| 0.5 | | $71.43 \pm 0.56$ | $74.17 \pm 0.91$ | $73.84 \pm 0.12$ | $73.66 \pm 0.88$ | $75.05 \pm 0.45$ |
| 1.0 | | $72.26 \pm 0.54$ | $74.46 \pm 0.44$ | $74.54 \pm 0.76$ | $74.91 \pm 0.42$ | $73.81 \pm 0.87$ |
| 2.0 | $71.23 \pm 0.71$ | $72.61 \pm 0.79$ | $73.68 \pm 0.17$ | $\mathbf{75.35 \pm 0.50}$ | $74.76 \pm 0.54$ | $74.46 \pm 0.56$ |
| 5.0 | | $72.60 \pm 0.45$ | $\mathbf{75.22 \pm 0.33}$ | $\mathbf{75.00 \pm 0.74}$ | $75.01 \pm 0.71$ | $73.96 \pm 1.60$ |
| 10.0 | | $72.29 \pm 0.48$ | $\mathbf{74.97 \pm 0.65}$ | $74.31 \pm 0.95$ | $74.03 \pm 0.30$ | $71.91 \pm 2.69$ |
| 100.0 | | $71.64 \pm 0.47$ | $72.92 \pm 0.39$ | $73.96 \pm 0.65$ | $73.13 \pm 0.73$ | $72.43 \pm 3.55$ |
| 1000.0 | | $71.58 \pm 0.53$ | $73.18 \pm 0.73$ | $73.32 \pm 1.70$ | $73.87 \pm 1.16$ | $72.52 \pm 1.92$ |
| $-$ | | $72.72 \pm 0.44$ | $\mathbf{75.22 \pm 0.19}$ | $\mathbf{75.05 \pm 0.42}$ | $72.51 \pm 3.89$ | $73.77 \pm 0.20$ |

Table 7: Ablation study of the parameters $\lambda$ and $\tau$ on the combination of {Path-OCT-Tissue}MNIST datasets with $N = 3$ participants over three independent runs. The first column ($\tau = 0.0$) corresponds to the FedAvg algorithm. The last row $(-)$ presents results for the `FedPeWS-Fixed` algorithm.

| $\lambda(\downarrow), \tau(\rightarrow)$ | $\tau = 0.0$ (FedAvg) | Proportion of warmup rounds $\tau = W/T$ | | | | |
|---|---|---|---|---|---|---|
| | | $\tau = 0.2$ | $\tau = 0.4$ | $\tau = 0.5$ | $\tau = 0.6$ | $\tau = 0.8$ |
| 0.0 | | $53.04 \pm 2.08$ | $50.35 \pm 0.61$ | $47.83 \pm 1.04$ | $47.89 \pm 1.17$ | $46.80 \pm 1.76$ |
| 0.1 | | $52.52 \pm 1.79$ | $51.72 \pm 0.63$ | $49.83 \pm 1.83$ | $48.83 \pm 2.51$ | $47.50 \pm 1.07$ |
| 0.3 | | $52.74 \pm 1.75$ | $54.36 \pm 0.80$ | $51.07 \pm 0.25$ | $50.94 \pm 2.22$ | $50.62 \pm 1.91$ |
| 0.5 | | $54.30 \pm 2.08$ | $54.42 \pm 1.43$ | $52.02 \pm 2.53$ | $51.43 \pm 1.60$ | $48.97 \pm 0.80$ |
| 1.0 | | $\mathbf{54.89 \pm 0.72}$ | $53.59 \pm 1.40$ | $51.49 \pm 1.19$ | $50.29 \pm 2.70$ | $52.21 \pm 1.02$ |
| 2.0 | $52.25 \pm 0.57$ | $\mathbf{54.75 \pm 1.12}$ | $\mathbf{54.45 \pm 1.78}$ | $53.02 \pm 0.45$ | $52.26 \pm 1.81$ | $52.49 \pm 0.49$ |
| 5.0 | | $\mathbf{54.91 \pm 0.91}$ | $\mathbf{54.99 \pm 0.90}$ | $52.41 \pm 0.45$ | $52.95 \pm 1.49$ | $52.53 \pm 1.35$ |
| 10.0 | | $\mathbf{55.12 \pm 1.16}$ | $\mathbf{55.03 \pm 1.39}$ | $52.85 \pm 0.95$ | $50.82 \pm 3.03$ | $52.27 \pm 1.11$ |
| 100.0 | | $54.22 \pm 1.74$ | $52.31 \pm 2.55$ | $52.10 \pm 0.87$ | $49.52 \pm 4.34$ | $51.46 \pm 2.92$ |
| 1000.0 | | $53.45 \pm 1.65$ | $53.82 \pm 2.16$ | $51.16 \pm 1.92$ | $51.30 \pm 2.02$ | $51.19 \pm 1.05$ |
| $-$ | | $53.69 \pm 0.77$ | $51.78 \pm 0.44$ | $50.24 \pm 1.88$ | $51.12 \pm 0.72$ | $49.87 \pm 1.23$ |

For the CIFAR-MNIST dataset, the preferred values of $\lambda$ that yield optimal outcomes range within $\{2.0, 5.0, 10.0\}$, and for $\tau$, the values are $\{0.4, 0.5\}$. A similar pattern is observed in the {Path-OCT-Tissue}MNIST dataset experiment, with a small difference, in which it shows a preference for fewer warmup rounds ($\tau \in \{0.2, 0.4\}$) and demonstrates optimal performance with the same set of $\lambda$ values. This consistency across different datasets indicates robustness in the parameter settings for achieving high accuracy.

### C.4 LARGE NUMBER OF PARTICIPANTS

Although our primary focus is on the cross-silo setting, we extend our study to include a large-scale scenario involving 200 participants on the CIFAR-MNIST dataset. We adopt a Dirichlet partition strategy with concentration parameter $\alpha = 0.5$ and implement this scenario with a partial participation rate of $0.1$. The outcomes of this experiment, as depicted in Figure 12, indicate a superior performance compared to the conventional FedAvg algorithm, thereby further substantiating the validity and effectiveness of our proposed method. The parameters set for `FedPeWS` are: $\tau = 25$ and $\lambda = 0.5$.

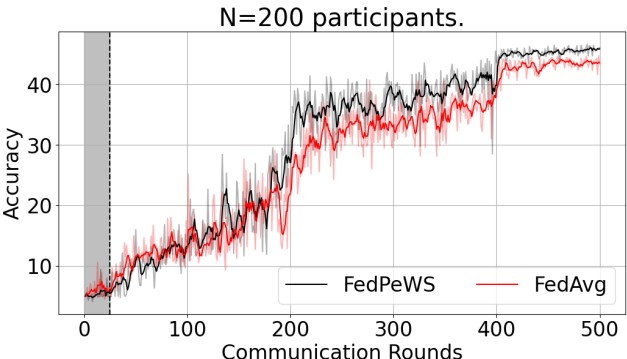

Figure 12: Visualization of global model performance with $N = 200$ participants with a partial participation rate of $0.1$. Smoothing is applied as a running average with a window size of 5. A learning rate scheduler is implemented at rounds 200 and 400 with a learning rate decay factor of $0.1$.

## D  NEURON ACTIVATIONS

In this section, we examine the extent to which neurons in each layer are activated. Our study uses the Synthetic-32K dataset and the `FedPeWS-Fixed` method (with parameter $= 50$). The vertical dashed line ($W = 50$) indicates the point at which participants switch to using full masks.

Figure 13 displays the neuron activations, measured as the sum of activations over a batch of samples randomly selected from each participant's dataset, over $T = 250$ communication rounds. The top row shows the outcomes for Participant 1, and the bottom row shows the activations for Participant 2. Each column corresponds to different fully connected layers (FC1 to FC4) in the network. Observations are as follows:

1. Before switching ($t \leq W$): for Participant $i$, subnetwork $i$ shows higher activation patterns in all given FC layers, $i \in [1, 2]$, while the other subnetwork exhibits a minimal activation.

2. After switching to full mask ($t > W$): (i) there is a noticeable increase in activations for both participants upon switching to using full masks, (ii) Participant 1 with its originally initialized subnetwork 1, shows a substantial increase in activations compared to subnetwork 2 across all layers. The same pattern is obesrved for Participant 2 with subnetwork 2.

These findings suggest that the personalized warmup strategy helps the network learn which paths to follow when specific data points are fed into the network. This supports the superiority of our method and corroborates the claims made in the main paper.

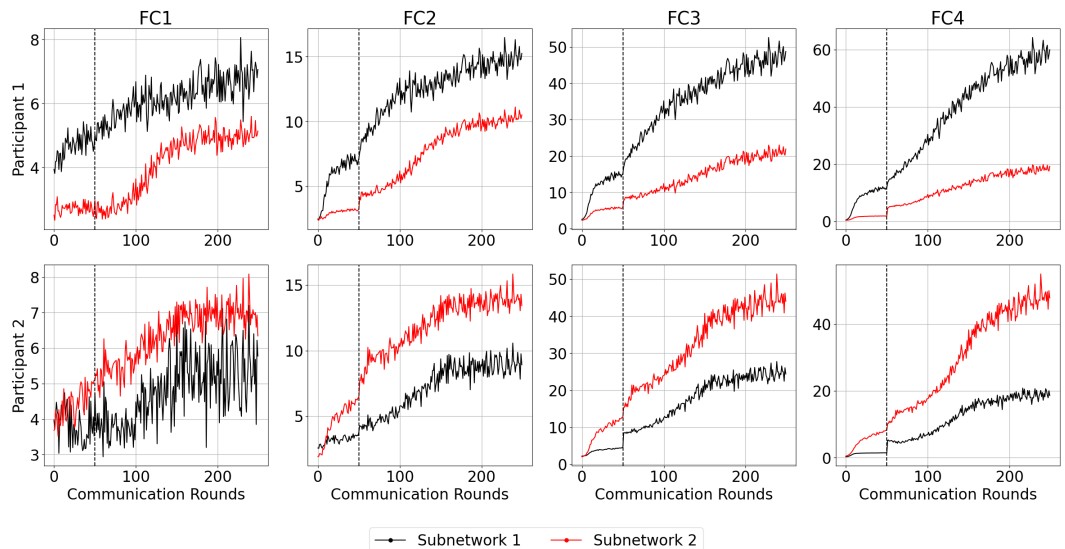

Figure 13: Neuron activation study on the Synthetic-32K dataset with a global learning rate $\eta_g = 1.0$. The experiment uses the FedPeWS-Fixed method with parameter $W = 50$, indicated by the vertical dashed line, marking the switch to full masks by each participant.

