# OpenReview forum: "FedPeWS: Personalized Warmup via Subnetworks for Enhanced Heterogeneous Federated Learning"
_ICLR.cc/2025/Conference — ICLR 2025 Conference Withdrawn Submission_

### Official Review · Reviewer_VcjV · 2024-10-31

**Soundness:** 2
**Presentation:** 3
**Contribution:** 2
**Rating:** 3
**Confidence:** 4

**Summary:**

This paper develops a warmup algorithm to tackle data heterogeneity in federated learning. The idea is to first allow clients to personalize their models in initial rounds, and then adopt conventional federated approach. Some experiments are provided to showcase the effectiveness of the algorithm. The paper is well written and easy to read.

**Strengths:**

- The idea of using independent subnet training in federated learning looks quite novel and reasonable to the reviewer.
- The paper is well written and easy to follow.

**Weaknesses:**

While the reviewer appreciates the novelty of the approach, they are a set of concerns/comments/questions:

- While the paper emphasizes that the primary benefit of the algorithm is improved convergence, there is a lack of theoretical analysis.
- How does the proposed method differ from FedPM (Isik et al., 2023) which also uses mask training?
- The biggest concern from the reviewer is that the numerical results are on a relatively simple setting:
   -- The datasets are too simple. The authors are suggested to try harder ones, e.g., CIFAR100, (Tiny) Imagenet
   -- The model used are too simple. The authors are suggested to try ResNet and ViT
   -- The comparison to SOTA is missing. Current comparison only involves FedAvg and FedProx and is far from enough. The authors are suggested to MOON, FedOpt, FedUV, etc.
   -- The number of clients is too small. While the authors tried N=200 in the appendix, the improvement is much less obvious than the simple setting in the main text.
   -- Important ablation study is missing. For example, the authors might try replacing the current warmup strategy by random neuron sampling and other IST approaches.
   -- Data heterogeneity with \alpha being 0.1 is not extreme. The authors are suggested to try smaller values, e.g., 0.01.

**Questions:**

See weaknesses

---

> ### Author Response · Authors · 2024-11-24
> **Response to Reviewer VcjV**
>
> We would like to thank the reviewer for taking their time to review our paper and noting that the presentation of the paper is good, and that the proposed approach is novel.
>
> > W1. Lack of theoretical analysis.
>
> While our work does not currently include a formal theoretical study, we would like to emphasize that our method is well-grounded in the context of federated learning (FL) and supported by extensive empirical evidence. To the best of our knowledge, there are no existing works in the FL literature that provide a theoretical analysis specifically for warmup-related topics. If the reviewer is aware of such work, we would greatly appreciate their guidance and will consider pursuing a theoretical investigation as part of our future work.
>
> Although a theoretical analysis is not included, our experiments consistently demonstrate positive gains with FedPeWS. Importantly, the worst-case performance matches that of the baseline algorithm (FedAvg), ensuring that our approach introduces no degradation. These empirical results validate the practical benefits of the proposed method.
>
> > W2. Difference from FedPM.
>
> While both approaches (our method and FedPM) involve mask training, the underlying goals and mechanisms are fundamentally different. FedPM focuses exclusively on improving communication efficiency by optimizing the bits-per-parameter (bpp) and trains solely on weight-level masks. In contrast, our approach addresses the challenges posed by statistical heterogeneity in federated learning. FedPeWS improves FL performance by training over both weights and masks, where the masks are applied at the neuron level. This design ultimately reduces the search space (please refer to Table 3 in the supplementary material which highlights these distinctions and comparisons to other methods).
>
> > W3.
>
> We thank the reviewer for their detailed and constructive feedback. Below, we address the concerns raised and provide clarifications:
>
> - **Datasets.** While we understand the reviewer’s suggestion to explore more complex datasets, we believe our current evaluation spans a diverse range of datasets, including CIFAR-10, MNIST, PathMNIST, OCTMNIST, TissueMNIST, and a synthetically generated dataset. These datasets cover different levels of complexity and heterogeneity. Additionally, we have conducted an experiment on CIFAR-100, where the dataset is split among $N=20$ clients, each assigned distinct 5 classes to emulate extreme heterogeneity. In this setup, our method demonstrates a significant improvement over the baseline algorithm, FedProx. The results are included [in this link](https://ibb.co/yVJkCkM).
>
> - **Models.** We acknowledge the importance of exploring deeper architectures like ResNet or ViT. However, applying our technique to models with residual connections introduces certain challenges that we leave for future work. We believe addressing these limitations in subsequent studies will extend the applicability of our method further.
>
> - **Baselines.** While FedAvg is included as a baseline, it is important to note that FedAvg is a special case of FedOpt. Regarding MOON, its principles conflict with our method, as MOON’s contrastive loss requires stable representations across time, which is incompatible with the dynamic subnetworks in our approach. Similarly, FedUV involves a large number of hyperparameters, making a direct comparison infeasible within the current scope.
>
> - **Number of clients.** In the supplementary material, we included an experiment with $N=200$ clients, demonstrating that our method retains its benefits in such scenarios. Additionally, we include a new CIFAR-100 experiment with $N=20$ clients to further validate our method under diverse client counts.
>
> - **Data heterogeneity.** While we used $\alpha = 0.1$ as the Dirichlet parameter for one of our data splits, generating splits with $\alpha = 0.01$ does not work, as some participants end up with no data (in the case with $N=10$). However, we already include experiments with non-overlapping class splits, which represent a highly heterogeneous scenario equivalent to $\alpha$ values smaller than 0.01. These setups highlight our method’s robustness under extreme heterogeneity.
>
>
> We hope these clarifications and additional experiments address the reviewer’s concerns.

---

> > ### Comment · Reviewer_VcjV · 2024-11-27
> > **Response**
> >
> > The reviewer appreciates the authors' efforts made for this response. While the reviewer acknowledges the additional results, I am afraid the current version is not ready yet for publication. Since this is an empirical paper without theoretical analysis, comparing the results with SOTA FL and PFL (in terms of convergence, accuracy, computation/communication overheads, etc) is necessary.
> >
> > Hence I would keep my score.

---

### Official Review · Reviewer_mkoE · 2024-11-02

**Soundness:** 1
**Presentation:** 2
**Contribution:** 1
**Rating:** 1
**Confidence:** 5

**Summary:**

The paper proposes "FedPeWS," a personalized warmup method for federated learning (FL) using subnetwork masking to improve model convergence under extreme data heterogeneity. It introduces a warmup phase where each participant trains a personalized subnetwork before switching to standard federated optimization, claiming enhanced accuracy and efficiency over baseline methods.

**Strengths:**

The paper studies an important problem which is convergence and performance issues of FL under extreme data heterogeneity.

**Weaknesses:**

Overall, while this paper attempts to address challenges in federated learning under extreme data heterogeneity, there are several critical issues regarding its hypotheses, methodology, and experimental validation. I hope these comments provide constructive insights for improvement:

* The paper is built upon unproven hypotheses, especially as stated in the abstract (lines 15-17): "we hypothesize that ... rounds." Presenting such assumptions without experimental or theoretical proof weakens the foundation of the study. For any claim regarding the benefits of a personalized warmup phase, empirical or theoretical evidence is essential to establish credibility.
* In lines 44-45, the authors suggest that the primary cause of federated learning failure under extreme data heterogeneity lies in high conflict between local updates during initial rounds. This premise is misleading, as conflicts due to objective misalignment and heterogeneity persist throughout the entire FL process, not just in early rounds. A more nuanced discussion around ongoing client conflicts across rounds would provide a clearer picture of challenges in FL.
* Lines 51-53 mention that each participant uses a personalized binary mask to learn local data distributions and optimize local (sparse) models. However, there is no clear explanation of how the nonlinear parameter space of a model can be mapped directly to each client’s data distribution. To the best of my knowledge, there is no work which has been able to drive a specific subnetwork structure representing the data distribution in NN with non-linear activation.
* The paper’s methodology seems to extend existing techniques without substantial innovation. Additionally, recent work in FL initialization with pre-trained models has shown that warmup rounds may not be necessary, as initialization with pre-trained or meta-learned models can often yield better results. A thorough discussion of and comparison with these approaches would better position the contributions within the current literature. I also urge the authors to enhance their literature review.
* The experimental setup lacks robust benchmarks, as it does not evaluate the method on deeper architectures or complex datasets like CIFAR-100. Additionally, comparisons with state-of-the-art FL methods (such as those involving advanced initialization or optimization techniques) would strengthen the empirical evaluation and actual benefits of the method. Right now, the actual benefits of method are questionable.

**Questions:**

See comments above.

---

> ### Author Response · Authors · 2024-11-24
> **Response to Reviewer mkoE (1/2)**
>
> > W1. W2.
>
> We appreciate the reviewer’s comments and concerns regarding the hypotheses presented in the paper. To address these points:
>
> For W1, we refer the reviewer to Appendix Section D, where we provide an empirical analysis of neuron activations for the FedPeWS-Fixed approach. This experiment was conducted to maintain controllability over specific subnetworks. Additionally, we include results for FedAvg (please refer to [this link](https://ibb.co/ZdWP8T1)), where no similar activation pattern is observed. In this case, data points from two different participants activate overlapping regions of the model, leading to slower convergence.
>
> The results in Appendix D empirically support our hypothesis mentioned in lines 15–17. Specifically, the personalized warmup phase facilitates a structured direction for the model to follow during training. For instance, data points from Participant 1 predominantly activate neurons in Subnetwork 1, even after the warmup period ends. Similarly, Participant 2's data points mainly activate neurons in Subnetwork 2, with minimal interference in Subnetwork 1. This behavior demonstrates the benefit of using a personalized warmup phase to guide convergence effectively.
>
> Regarding W2, while we agree that conflicts due to objective misalignment and heterogeneity can persist throughout the entire FL process, our analysis focuses on the early rounds, where the lack of proper initialization amplifies these conflicts. By addressing this critical phase through the personalized warmup, FedPeWS provides a structured initialization that mitigates the early-stage conflicts, as supported by the neuron activation analysis in Appendix D. We will revise the manuscript to clarify this distinction and explicitly refer readers to the empirical evidence provided in Appendix D.
>
> > W3.
>
> To clarify, our work does not aim to construct an exact subnetwork representing a specific data distribution. Instead, the objective is to find subnetworks that adapt to the degree of heterogeneity in the data. For example, in scenarios of extreme heterogeneity, one would expect the subnetworks (or neurons) assigned to different participants to have minimal or no overlap, allowing each participant to focus on its unique data distribution.
>
> Regarding the reviewer’s comment about the lack of prior work addressing this, we would like to highlight the research around the Lottery Ticket Hypothesis (LTH). LTH investigates the existence of sparse subnetworks (winning tickets) that are highly effective for specific tasks and are conceptually related to aligning subnetworks with distinct data distributions. While our work does not directly follow LTH, it is inspired by similar ideas of identifying effective subnetworks. We would appreciate the reviewer’s perspective on whether this aligns with their understanding of LTH.
>
> > W4.
>
> We appreciate the reviewer’s comment and the suggestion to position our contributions more clearly within the current literature. However, we would like to clarify that our proposed FedPeWS method is completely orthogonal to FL initialization with pre-trained or meta-learned models and addresses a fundamentally different objective. Even with a better initialization, FL convergence will be affected if the local data distributions of the participants are extremely heterogeneous.
>
> FedPeWS focuses on mitigating the challenges of extreme data heterogeneity by dynamically tailoring subnetworks during the warmup phase. This phase is not intended to simply initialize the global model but rather to guide the model towards a structured convergence path that aligns better with participant-specific data distributions. As such, FedPeWS and pre-trained or meta-learned initialization methods address distinct challenges and are not directly comparable.
>
> We agree that incorporating a more thorough discussion of related work in FL initialization and meta-learning techniques would enrich the manuscript. We will update the literature review to include these perspectives and clarify the differences between FedPeWS and initialization-based approaches.

---

> ### Author Response · Authors · 2024-11-24
> **Response to Reviewer mkoE (2/2)**
>
> > W5.
>
> We appreciate the reviewer’s concern regarding the experimental setup and the call for additional benchmarks.
>
> To address this, we include an additional experiment on the FedProx method, conducted with $N=20$ participants on the CIFAR-100 dataset. The data partitioning strategy follows a class-based split, where each participant is randomly assigned a distinct set of 5 classes without any overlap with other participants. This setup emulates a scenario of extreme heterogeneity. The results of this experiment are presented via [this link](https://ibb.co/yVJkCkM), further demonstrating the robustness of our method under complex conditions.
>
> Additionally, Figures 6a and 6b already evaluate our method on diverse datasets, including CIFAR-10, MNIST, and medical imaging datasets such as PathMNIST, OCTMNIST, and TissueMNIST. These datasets encompass varying levels of complexity and heterogeneity, making them highly transferable to other datasets in terms of benchmarking.
>
> Furthermore, Figure 5 demonstrates that even in simpler scenarios, such as a class-based split with extreme heterogeneity, FedAvg struggles to achieve satisfactory results. In contrast, FedPeWS shows a significant performance improvement, underscoring its ability to handle heterogeneity effectively.
>
> We would also like to emphasize that our work introduces a foundational approach to addressing a critical gap in federated learning under extreme heterogeneity. While it incorporates some existing techniques (e.g., mask training), FedPeWS represents a novel and innovative idea aimed at solving a distinct challenge. We believe this uniqueness should not be overshadowed by overlaps with specific existing techniques, as the novelty lies in the overall conceptual framework and mechanisms of the personalized warmup phase.
>
> We hope this additional evidence and clarification will convince the reviewer of the substantial contributions and encourage them to reassess their evaluation of our work.

---

> > ### Comment · Reviewer_mkoE · 2024-11-27
> > **Response to authors**
> >
> > Thank you for your response. However, my concerns remained unresolved. See my feedback below:
> >
> > > Regarding W2, while we agree that conflicts due to objective misalignment and heterogeneity can persist throughout the entire FL process, our analysis focuses on the early rounds, where the lack of proper initialization amplifies these conflicts.
> >
> > Again, the conflicts exist the during the entire FL due to the data heterogeneity nature. And, the whole motivation of paper suggesting warmup rounds solving this issue is questionable. Also, there are studies using pre-trained weights as initialization, no need for warmups. In order to prove a claim authors need to provide enough experiments.
> >
> > > For example, in scenarios of extreme heterogeneity, one would expect the subnetworks (or neurons) assigned to different participants to have minimal or no overlap, allowing each participant to focus on its unique data distribution.
> >
> > I disagree with this. Overlap or no overlap in functional space does not necessitate overlap or no overlap in the parameter space. This needs rigorous experimentation or theory to be proven. Also, LTH does not support this claim "aligning subnetworks with distinct data distributions".
> >
> > > However, we would like to clarify that our proposed FedPeWS method is completely orthogonal to FL initialization with pre-trained or meta-learned models and addresses a fundamentally different objective.
> >
> > I disagree. The whole motivation of your study in proposing the warmup rounds is to achieve a better initialization as you conjecture this is the bottleneck in data heterogeneous FL.
> >
> > > Even with a better initialization, FL convergence will be affected if the local data distributions of the participants are extremely heterogeneous.
> >
> > Then you just disapprove the whole premise of your paper.
> >
> > Overall, I believe the whole premise of the paper is questionable, and not technically correct. Therefore, I keep my score.

---

### Official Review · Reviewer_2YSy · 2024-11-05

**Soundness:** 2
**Presentation:** 2
**Contribution:** 2
**Rating:** 3
**Confidence:** 4

**Summary:**

To address the problem of extreme data heterogeneity in federated learning, this paper starts from the perspective of initialization. It argues that an appropriate personalized initial method can enable the clients to quickly learn their local data well before engaging in broader collaboration, and thus achieve faster convergence and a higher final accuracy. In this paper, the authors propose FedPeWS, a personalized warm-up method for the initial phase of federated learning training, which simultaneously optimize the personalized masks of the clients and update the corresponding sub-networks during the initial warm-up phase to enables the global network to adapt faster and better to extreme data heterogeneity scenarios. Experimental results under various scenarios and parameter settings validate the effectiveness of this method.

**Strengths:**

1. This paper addresses the problem of data heterogeneity from a novel perspective.
2. The authors have conducted extensive experiments under a variety of datasets and made a detailed analysis of the ablation experiments around the relevant parameters.

**Weaknesses:**

1. As mentioned by the authors, this paper lacks proofs related to convergence, and therefore at the same time, it fails to give an analysis of the relationship between hyperparameters such as $\lambda$, $\tau$ and convergence to determine a better range of theoretical values. Therefore for different experimental scenarios, perhaps multiple experiments (e.g. grid search) are required to determine the most suitable parameter values, which has limitations in terms of generalizability and cost.
2. The experimental results show that the effectiveness of FedPeWS and FedPeWS-Fixed methods are comparable, and in some stages FedPeWS-Fixed even outperforms FedPeWS, which I think is an issue that should not be ignored. In conjunction with Appendix B.2, I think FedPeWS-Fixed is a rigid approach to divide the network evenly and fixedly, while the FedPeWS method dynamically optimizes the personalized masks to select the appropriate sub-networks. Intuitively this dynamic optimization method should be more effective than the simple fixed division method, but the experimental results are the opposite, does it mean that the dynamic optimization method of FedPeWS does not achieve much extra gain? I think this is a question worth analyzing.

**Questions:**

1. Since the research purpose of this paper is to solve the problem of extreme data heterogeneity, more comparisons can be made with some of the current state-of-the-art methods for solving data heterogeneity. I think that besides the FedProx method mentioned in this paper, the MOON [1] method is also a worthy comparison. And I think FedPeWS is essentially an initialization method, so it can also be combined with methods such as PFL [2] in the subsequent standard federated optimization to investigate whether this method, FedPeWS, can bring further enhancement to the existing state-of-the-art methods.
2. In the related experimental descriptions of Table 1 and Table 2, there is no mention of the number setting of clients N. If I understand correctly, the settings on synthetic datasets for both experiments are N=2. This implies that when comparing with FedProx, the comparison has been made only in the scenario of synthetic datasets (N=2). If possible, I'd like to see experimental comparisons with the FedProx method under more datasets (and at the same time larger N value).
3. In this paper, collaboration rounds are used in the experimental part of the main text to compare the convergence speeds of FedAvg and FedPeWS, but since FedPeWS contains a greater computational overhead for one collaboration round during the warm-up phase (but only need to transmit sub-network, which is theoretically less expensive in terms of communication overheads), simply comparing the number of rounds is not convincing enough. Although the experimental analysis for wall-clock time is also presented in the Appendix, it can also be seen that FedPeWS does take more time in the warm-up phase, and thus the balance between the parameter $\tau$ and the actual convergence speed is very important in practical applications. To summarize, I think how to determine the size of the warm-up rounds $W$, or the size of $\tau$, in the experiments is an issue worth discussing, which also involves the analysis of the overall computational and communication overheads. (By the way, if I understand correctly, $\tau =25$ in Appendix C.4 should be changed to $W=25$)

[1] Li, Qinbin, Bingsheng He, and Dawn Song. "Model-contrastive federated learning." Proceedings of the IEEE/CVF conference on computer vision and pattern recognition. 2021.
[2] Arivazhagan, Manoj Ghuhan, et al. "Federated learning with personalization layers." arXiv preprint arXiv:1912.00818 (2019).

---

> ### Author Response · Authors · 2024-11-24
> **Response to Reviewer 2YSy (1/2)**
>
> We sincerely thank the reviewer for taking the time to review our paper and for acknowledging the novelty of our approach, as well as the thoroughness of our experiments and detailed analyses.
>
> > W1. Relationship between $\lambda$ and $\tau$.
>
> We would like to refer the reviewer to Figure 6, particularly the heatmap plots, which illustrate the impact of $\lambda$ and $\tau$ on the final performance after convergence. The results in Figure 6 indicate that the optimal values for $\lambda$ fall within the set $\{5, 10\}$, while $\tau = 0.4$ yields the best performance. This is true for both the datasets (CIFAR-MNIST and {Path-OCT-Tissue}MNIST), which correspond to very different domains. Therefore, an exhaustive search for these parameters may not always be necessary. However, we do recognize that some hyperparameter tuning may be required for completely new problem domains.
>
> > W2. The effectiveness of FedPeWS and FedPeWS-Fixed methods are comparable.
>
> We agree that FedPeWS and FedPeWS-Fixed methods yield comparable performance under some settings, especially when there are only $N = 2$ participants. We would like to emphasize again that the key insight driving our method is that extreme heterogeneity hampers convergence due to conflicts between the updates in the initial collaboration rounds. By personalizing the warmup phase, our method minimizes these conflicts and facilitates faster convergence to a better solution under large heterogeneity. In FedPeWS-Fixed, there is no overlap between the subnetworks, and hence, there are zero conflicts during the warmup phase. This explains the good performance of FedPeWS-Fixed under some settings (see Figure 6a) with $N=2$ participants. However, it should be obvious that FedPeWS-Fixed is not a scalable approach as the number of participants ($N$) increases. As $N$ increases, the size of each subnetwork becomes smaller leading to suboptimal performance, especially when the tasks to be learned are more complex. This is evident from Figure 6b, where FedPeWS clearly outperforms FedPeWS-Fixed when tested on complex medical datasets, highlighting the limitations of the fixed partition approach. Thus, FedPeWS-Fixed is not a practical solution when the data distributions are not known apriori and the number of participants is not small. FedPeWS is the recommended method and its adaptive design ensures it can handle a wide range of complexities and consistently outperforms FedPeWS-Fixed in most cases.

---

> ### Author Response · Authors · 2024-11-24
> **Response to Reviewer 2YSy (2/2)**
>
> > Q1.
>
> Thank you for your insightful comments. Our primary goal in this work is to address federated learning settings where a good global model exists, but learning it becomes challenging due to extreme data heterogeneity. We show that FedPeWS effectively tackles this issue by minimizing the update conflicts during the initial communication rounds and facilitating convergence to a better solution, as demonstrated in combination with FedAvg and FedProx.
>
> We agree that extending FedPeWS to personalization-focused methods such as PFL [2] or other state-of-the-art techniques like MOON [1] is a promising avenue for future work. While our results strongly suggest that FedPeWS can provide similar benefits when combined with these methods, exploring such extensions is beyond the scope of our current research. Nonetheless, we appreciate your suggestion and believe it opens up exciting possibilities for further enhancing the impact of FedPeWS in diverse federated learning paradigms.
>
> > Q2.
>
> We include an additional experiment on the FedProx method, conducted with $N=20$ participants on the CIFAR-100 dataset. The data partitioning strategy follows a class-based split, where each participant is randomly assigned a distinct set of 5 classes without any overlap with other participants and $\lambda=0.0$. This setup emulates a scenario of extreme heterogeneity. The results of this experiment can be found using [this link](https://ibb.co/yVJkCkM).
>
> > Q3.
>
> We appreciate the reviewer’s insightful comment regarding the comparison of convergence speeds and the computational overhead of FedPeWS during the warmup phase.
>
> To address the concern, we note that while FedPeWS does introduce a slight increase in wall-clock time compared to the baseline algorithm during the warmup phase, this increase is marginal rather than substantial (e.g., it is not $2\times$ or more). Furthermore, the enhanced convergence performance and improved results in heterogeneous scenarios justify the additional computational effort.
>
> As for the balance between the warmup round size ($W$) and the overall convergence speed, our experimental results indicate that tuning $W$ appropriately can significantly improve performance without introducing excessive overhead. Additionally, while the warmup phase requires slightly more computation, the communication cost during this phase is reduced due to the transmission of subnetwork parameters instead of the entire model.
>
> We also appreciate the typo suggestion regarding Appendix C.4, and we will correct the notation from $\tau$ to $W$ to ensure consistency and clarity.

---

### Official Review · Reviewer_zTLf · 2024-11-07

**Soundness:** 2
**Presentation:** 3
**Contribution:** 1
**Rating:** 3
**Confidence:** 4

**Summary:**

The paper proposes a personalized federated learning algorithm called FedPeWS. The paper proposes to split the learning rounds into two parts. The first $W$ rounds is called warmup phase where each client learns a personalized subnetwork of the model. The key parts of the proposed algorithm is to identify the subnetwork for each client and determining the warmup length.

**Strengths:**

The paper studies the problem of personalized federated learning from learning subnetworks point of view. The presentation of the paper is good.

**Weaknesses:**

- The proposed algorithm is too intuitive. And reading the paper I am not clear why the proposed algorithm can help personalized federated learning. Also reading the introduction, the motivation is not convincing to me.
- Since this work is based on intuitions, I believe the paper should improve the experimental study significantly. Specifically, I believe the paper should add more baselines to the paper with more analytical explanations about the result.

**Questions:**

Based on my understating, in the first phase each client learns its own subnetwork. Then in the second phase the proposed algorithm employs federated averaging. Can you explain why does the algorithm split the learning rounds into two parts? I am not clear about the intuition behind this and its helpfulness in personalized federated learning.

---

> ### Author Response · Authors · 2024-11-24
> **Response to Reviewer zTLf**
>
> We sincerely thank the reviewer for taking the time to review our paper and for acknowledging that the presentation of the paper is good. However, we would like to clarify a key misunderstanding regarding our work. While the reviewer described FedPeWS as a personalized federated learning (PFL) algorithm, we respectfully disagree with this characterization. Unlike PFL, all the clients in FedPeWS learn a single global model at the end of the collaboration. Our method employs ``personalization'' (learning of tailored subnetworks for each participant) ***only*** during the warmup phase to handle extreme statistical heterogeneity. Hence, our method cannot be compared to existing PFL methods.
>
> > W1: The proposed algorithm is too intuitive. And reading the paper I am not clear why the proposed algorithm can help personalized federated learning. Also reading the introduction, the motivation is not convincing to me.
>
> Once again, we would like to emphasize that our method is not designed for personalized federated learning (PFL), nor does it help PFL. It is a method to better handle heterogeneity in a traditional cross-silo federated learning setting. The key insight driving our method is that extreme heterogeneity hampers convergence due to conflicts between the updates in the initial collaboration rounds. By personalizing the warmup phase, our method minimizes these conflicts and facilitates faster convergence to a better solution under large heterogeneity. Our method does not offer any benefit under IID setting and performs comparably to FedAvg when the local data distributions are IID.
>
> > W2: Since this work is based on intuitions, I believe the paper should improve the experimental study significantly. Specifically, I believe the paper should add more baselines to the paper with more analytical explanations about the result.
>
> Yes, our method is based on empirical observations. We also acknowledge in the Conclusions section that the lack of theoretical convergence analysis is a limitation of the current work. We would greatly appreciate if the reviewer could provide specific suggestions on appropriate baselines to improve the experimental study. To the best of our knowledge, personalized warmup is a novel concept that has not been studied in the existing literature. We have provided more analytical explanations in the supplementary material including a study on neuron activations.

---

### Note · Authors · 2024-12-03

I have read and agree with the venue's withdrawal policy on behalf of myself and my co-authors.